# The Prognostic Value and Clinical Utility of the 40-Gene Expression Profile (40-GEP) Test in Cutaneous Squamous Cell Carcinoma: Systematic Review and Meta-Analysis

**DOI:** 10.3390/cancers15092456

**Published:** 2023-04-25

**Authors:** Razan Masarwy, Shahaf Shilo, Narin Nard Carmel Neiderman, Liyona Kampel, Gilad Horowitz, Nidal Muhanna, Jobran Mansour

**Affiliations:** 1Department of Otolaryngology, Head and Neck and Maxillofacial Surgery, Tel-Aviv Medical Center, Tel-Aviv University, Tel-Aviv 6423906, Israel; 2Sackler Faculty of Medicine, Tel Aviv University, Tel Aviv 6997801, Israel

**Keywords:** 40-gene expression profile (40-GEP), cutaneous squamous cell carcinoma, skin squamous cell carcinoma, metastasis

## Abstract

**Simple Summary:**

A subset of cutaneous squamous cell carcinoma (cSCC) that is associated with high-risk features carries an increased risk for aggressive disease, local recurrence, and regional metastasis, which leads to significant morbidity and mortality. The current available systems for cSCC risk stratification are considered inadequate and fail to accurately distinguish high-risk cSCC patients who can benefit from aggressive management, emphasizing a great need for an accurate metastatic risk tool for early identification of high-risk cSCC patients. A 40-gene expression profile (40-GEP) was developed to biologically distinguish high-risk tumors that would benefit from more aggressive management. This systematic review and meta-analysis aimed to evaluate the reported prognostic value and clinical utility of the 40-GEP test. The results indicated that the 40-GEP test improved cSCC risk stratification both independently and in combination with other clinicopathologic risk factors.

**Abstract:**

Background: The current tumor staging systems for cutaneous squamous cell carcinoma (cSCC) are considered inadequate and insufficient for evaluating the risk of metastasis and for identifying patients at high risk of cSCC. This meta-analysis aimed to assess the prognostic significance of a 40-gene expression profile (40-GEP) both independently and integrated with clinicopathologic risk factors and established staging systems (American Joint Committee on Cancer, eighth edition (AJCC8) and Brigham and Women’s Hospital (BWH)). Methods: Electronic databases, including PubMed (MEDLINE), Embase, the Cochrane Library, and Google Scholar, were systematically searched to identify cohort studies and randomized controlled trials on evaluations of the prediction value of 40-GEP in cSCC patients up to January 2023. The metastatic risk analysis of a given 40-GEP class combined with tumor stage and/or other clinicopathologic risk factors was based upon log hazard ratios (HRs) and their standard error (SE). Heterogeneity and subgroup analyses were performed, and data quality was assessed. Results: A total of 1019 patients from three cohort studies were included in this meta-analysis. The overall three-year metastatic-free survival rates were 92.4%, 78.9%, and 45.4% for class 1 (low risk), class 2A (Intermediate risk), and class 2B (high risk) 40-GEP, respectively, indicating a significant variation in survival rates between the risk classification groups. The pooled positive predictive value was significantly higher in class 2B when compared to AJCC8 or BWH. The subgroup analyses demonstrated significant superiority of integrating 40-GEP with clinicopathologic risk factors or AJCC8/BWH, especially for class 2B patients. Conclusions: The integration of 40-GEP with staging systems can improve the identification of cSCC patients at high risk of metastasis, potentially leading to improved care and outcomes, especially in the high-risk class 2B group.

## 1. Introduction 

Cutaneous squamous cell carcinoma (cSCC) is the second most common cancer in the United States. It is estimated that the overall incidence of cSCC increased by 263% between 1976–1984 and 2000–2010. Although the majority of early-stage tumors can be successfully cured with surgical excision, a subset of cSCCs is associated with high-risk features and carries an increased risk of local recurrence and regional metastasis with significant morbidity and mortality [1,2,3]. The optimal management of high-risk node-negative head and neck cSCC remains controversial and has yet to be established.

Therapeutic options are currently available for locally advanced cSCC, but means to accurately and promptly identify patients with a higher risk of metastasis at an early stage are lacking. Their application could vastly facilitate a more aggressive treatment with lymph node dissection and potentially a timely adjuvant therapeutic intervention, ultimately lowering the risk of recurrence and metastasis. In addition, tailoring treatment plans of patients identified as being at high risk can also prevent the performance of costly and invasive procedures for low-risk patients.

The National Comprehensive Cancer Network (NCCN) guidelines stratify cSCC patients into a high-risk group according to clinicopathological features that are associated with local recurrence and metastasis. Poor differentiation, perineural involvement, lymphatic or vascular involvement, and invasion beyond subcutaneous fat have all demonstrated a greater risk of metastasis and are therefore indications that more aggressive management is required [4,5,6]. The American Joint Committee on Cancer (AJCC) cancer staging manual 8th edition (AJCC8) and Brigham Women’s Hospital (BWH) are both tumor (T) staging systems that translate clinical and pathological features into tumor stages that correlate with poor outcomes [7,8]. The BWH staging system is simple, consisting of just four variables, such as perineural invasion, with a number of prognostic factors categorizing the risks. However, this system depends on the documentation of the diameter of the invaded nerve (<0.1 mm or >0.1 mm), which is not consistently reported by pathologists who examine primary malignant tumors. Nonetheless, although the AJCC 8 staging system is a significant update from the AJCC 7 system, the update only covers head and neck cSCCs, limiting its practicality [9,10].

BWH and AJCC8’s accuracies for predicting metastatic risk, however, are relatively low (i.e., low positive predictive value (PPV) for metastasis), with 30% of metastatic cases misclassified as low T stage and more than 70% of those classified as high-risk T stage not developing metastasis [9,11]. This underscores the importance of exploring alternative stratification systems that can accurately classify patients based upon their risk and ultimately enable the early identification of primary cSCC patients at high risk of developing metastases.

Roscher et al. evaluated the predictive capacity of the current staging systems, including the American Joint Committee on Cancer (AJCC) 7th and 8th editions and the Breuninger and Brigham and Women’s Hospital (BWH) staging systems, for cSCC. The study findings revealed that these four staging systems had limited ability to anticipate the development of metastatic disease and had poor-to-moderate discriminatory power in distinguishing between patients who experienced metastasis and those who did not. The poorest results were found for the AJCC 7 system, which was the most widely used staging system in the study. Although the AJCC 7 system provided an increased risk of metastasis for T2 patients as compared with T1 patients, none of the patients had an advanced stage (T3 and T4 categories). In contrast, the BWH system and, to a lesser degree, the Breuninger system yielded an elevated risk of metastasis by increasing stage or risk category and delivered slightly better results than those of the AJCC 7 or AJCC 8 systems. Consequently, the study’s principal finding was that these four staging systems were deemed unsatisfactory for clinical use [10].

Tumor diameter and thickness, PNI (perineural invasion), deep invasion beyond the subcutaneous fat, poor differentiation, and immunosuppression are well-known risk factors for aggressive cSCC and are often used to identify tumors with a high-risk of metastasis. A comprehensive meta-analysis has investigated the factors linked to worse outcomes of cSCC. The study found that tumor depth is strongly associated with increased risk of metastasis. Additionally, the analysis identified other risk factors, including tumor diameter exceeding 20 mm, poor differentiation, and PNI, which were also associated with increased risk of metastasis [6]. On the other hand, Ibrahim et al. found a limited HR for metastasis associated with each clinicopathological factor when compared with gene profiling [12].

The 40-gene expression profile (40-GEP) molecular test was developed using archival formalin-fixed paraffin-embedded (FFPE) primary cSCC to predict the metastatic risk based upon the biology of the tumor [13,14]. An assay of 40 genes associated with metastatic cSCC was created using deep machine learning, and it has been validated for the provision of prognostic estimates by classifying patients into being at low risk (class 1), moderate risk (class 2A), or high risk (class 2B) of sustaining recurrence or metastasis [12,13,15,16].

Better prediction of metastatic risk was achieved with 40-GEP compared to the AJCC8 and BWH staging systems, however, it has not yet been incorporated into the NCCN guideline recommendations. The purposes of this meta-analysis were to assess the predictive value of 40-GEP in the setting of cSCC as an independent tool and in combination with other staging systems and clinicopathological risk factors in identifying high-risk tumors with elevated risk of metastasis that might benefit from addressing the lymph node basin and aggressive management.

## 2. Material and Methods

This systematic review followed the Preferred Reporting Items for Systematic Reviews and Meta-Analyses (PRISM, 2009) framework guidelines and the Meta-Analysis of Observational Studies in Epidemiology guidelines [17,18]. (Appendix A) The detailed protocol is documented online in the International Prospective Register of Systematic Reviews registry (PROSPERO) (CRD42023400695). No approval was required from the institutional review board nor the ethical committee according to local law because it does not use any individualized patient data.

### 2.1. Search Strategy

We conducted systematic manual searches in PubMed (MEDLINE), EMBASE, the Cochrane Library, and Google Scholar to identify all randomized clinical trials, prospective observational studies, and retrospective cohort studies that evaluated the prognostic efficacy of 40-GEP in cSCC that were published since January 2023. A predefined search algorithm was used using the following keywords: 40-gene expression profile (40-GEP), gene expression analysis, cutaneous squamous cell carcinoma, skin squamous cell carcinoma, metastasis, and metastatic risk (Appendix A). The references lists of all relevant studies were checked for additional references. No language or date restrictions were applied.

### 2.2. Study Selection

The inclusion criteria were as follows: (1) FFPE samples of primary cSCC, (2) subjects with no evidence of regional or distant metastasis, (3) subjects with fewer than 3 years of documented follow-up, and (4) patients with one or more high-risk features as defined by the NCCN guidelines. Excluded were animal and in vitro studies, studies with incomplete data, duplicate publications, case reports, case series, review articles, and guidelines.

### 2.3. Data Extraction

Two investigators independently identified and extracted articles for potential inclusion. Disagreements were resolved by referral to a third reviewer. The full texts of the remaining references were then retrieved and analyzed. The primary endpoints were three-year metastasis-free survival (MFS) including regional and distant metastasis. Accuracy metrics that included sensitivity, specificity and positive and negative predictive values (PPV and NPV) were also assessed and compared for 40-GEP class 2 (2A/2B), AJCC8 (T3/T4) and BWH (T2b/T3) to determine significant variances in the accuracy of risk stratification of metastasis development.

### 2.4. Quality Assessment and Risk of Bias

We analyzed quality and risk of bias with the Newcastle-Ottawa Quality Assessment Scale (NOS) for assessing the quality of nonrandomized studies [19]. The scale is based upon eight criteria and provides a score ranging from no stars (high risk of bias) to nine stars (low risk of bias). We determined the risk of bias for each study and designated a rating of five stars or below as representing high risk, six to seven stars as intermediate risk, and eight to nine stars as low risk. The assessment was carried out independently by two investigators. A summary assessment of the risk of bias (high, intermediate, or low) was derived for each outcome in each study (Appendix A).

### 2.5. Data Analysis

Statistical analyses were performed using RStudio (R Foundation for Statistical Computing) (v3.6.3) to generate prediction intervals and forest plots of MFS and accuracy metrics rates. Inverse-variance, random-effect pooled hazard ratios (HRs) were calculated with the corresponding 95% confidence intervals (CIs) using RevMan version 5.4 (The Nordic Cochrane Centre, the Cochrane Collaboration) to summarize the overall and within subgroup results for metastatic risk of combining 40-GEP with class 2 (2A/2B) with clinicopathologic risk factors or with AJCC8/BWH staging systems. A *p*-value of <0.05 was considered statistically significant. Result heterogeneity among the studies was quantified using the heterogeneity index (I^2^), and a value higher than 50% was considered as being substantial heterogeneity.

## 3. Results

### 3.1. Study Selection

Figure 1 is a flow diagram of publication retrieval and selection. The search strategy identified 446 citations, including 37 abstracts that met the initial screening criteria. After reviewing the full-length articles, three studies met all of the inclusion criteria for the meta-analysis, with a total of 1019 patients. The study conducted by Wysong et al. included a cohort of individuals with 73.2% men, a median age of 70 years old (ranging from 34 years old to 95 years old), and 66.7% of them had cSCC located in the head and neck region. A similar pattern was observed in the study by Ibrahim et al., where 73.3% of the participants were men, the median age was 71 years old (ranging from 34 years old to 95 years old), and 66.2% had cSCC located in the head and neck region. A third study, by Arron et al., had 82.4% men, and all of them had cSCC located in the head and neck region (Table 1). All three cohort studies were published between 2020 and 2022 [12,13,15]. We performed a quality assessment for risk of bias using the NOS, and all three studies had scores above five, making them eligible for inclusion (Appendix A).

### 3.2. Metastasis-Free Survival (MFS) Rates

Figure 2 forest plot of the 3-year-MFS of 40-GEP. (Figure 2A) 3-year MFS 40-GEP (class 1) rate, (Figure 2B) 3-year MFS 40-GEP (class 2A) rate, (Figure 2C) 3-year MFS 40-GEP (class 2B) rate.

A total of 541 cases were classified as 40-GEP class 1, 412 cases were classified as class 2A, and 66 cases were classified as class 2B (Figure 2). The 3-year MFS rates as derived from Kaplan–Meier analyses were 92.52% (95%CI 89.96–94.46%), 78.21% (95%CI 74.57–81.45%), and 45.16% (95%CI 33.64–57.22%), respectively.

### 3.3. Accuracy Metric Comparison for Cutaneous Squamous Cell Carcinoma per 40-GEP Class and AJCC8/BWH T Stage

The accuracy of risk prediction for the 40-gene expression profile class, the Brigham and Women’s Hospital, and the American Joint Committee on Cancer Cancer Staging Manual, Eighth Edition binary T stage is show in Figure 3.

A comparison of the accuracy metrics with AJCC8 T3/T4 and BWH T2b/T3 revealed that the overall 40-GEP class 2B from the three included studies showed a higher PPV of 55.9% (95%CI 50.22–61.44) (Figure 3A) compared to 33.3% (95%CI 28.21–38.87) and 36.7% (95%CI 31.11–41.98) for high-stage AJCC8 and BWH, respectively (Figure 3C,D). PPV for the GEP-40 groups (class 2A/2B) was 26.9% (95%CI 22.24–32.29) compared to 33% (95%CI 28.21–38.87), and 36.3% (95%CI 31.11–41.98) for AJCC8 and BHW, respectively. The overall NPV results of 40-GEP class 2B and class 2 (2A/2B) were 85.8% (95%CI 81.37–89.34) and 92.4% (95%CI 88.87–94.97), respectively. The NPV for AJCC8 and BWH were 86.92% (95%CI 82.59–90.30) and 85.99% (95%CI 81.57–89.49), respectively. (Figure 3E–H). The overall sensitivity results of 40-GEP class 2B and class 2 (2A/2B) were 22.10% (95%CI 16.19–29.39) and 74.45% (95%CI 64.74–82.21), respectively. The sensitivity for AJCC8 and BWH were 37.87% (95%CI 32.55–43.49) and 28.33% (95%CI 23.51–33.70), respectively. (Appendix A). The overall specificity results of 40-GEP class 2B and class 2 (2A/2B) were 96.52% (95%CI 93.73–98.10) and 58.62% (95%CI 47.99–68.49), respectively. The specificity for AJCC8 and BWH were 84.63% (95%CI 80.09–88.29) and 89.97% (95%CI 86.02–92.90), respectively (Appendix A).

### 3.4. 40-GEP Prognostic Accuracy Combined with Other Staging Systems (AJCC8 and BWH Tumor Classification)

We integrated the 40-GEP test results (class 2A or 2B) with clinicopathologic factors risk and tumor staging systems (AJCC8/BWH) in order to determine the complementary effect of 40-GEP on existing risk assessment strategies. The pooled overall HRs from the multivariate analysis for the combined GEP class 2B and high-stage AJCC8 (T3/T4) and BWH (T2b/T3) were 9.98 (95%CI 6.41–15.52) and 8.62 (95%CI 5.48–13.55), respectively. The pooled overall HRs from the multivariate analysis for combined 40-GEP class 2A and AJCC8 or BWH were 2.65 (95%CI 1.82–3.85) and 2.69 (95%CI 1.85–3.91), respectively. The pooled overall HRs for AJCC8 and BWH alone were 2.61 (95%CI 1.94–3.52) and 2.18 (95%CI 1.55–3.07), respectively. The addition of 40-GEP class 2B to either AJCC8 or BHW demonstrated a more than 3–4-fold greater prognostic value compared to the combined class 2A/(AJCC/BWH), AJCC8, and BWH staging alone (Figure 4A,B).

Similarly, the pooled overall HR from the combination of 40-GEP class 2B with clinicopathological risk factors (tumor thickness, tumor diameter, poor differentiation, PNI, invasion beyond subcutaneous fat) yielded a significantly improved HR (6.66, 4.04–10.96) when compared to the pooled HR from the combination of 40-GEP class 2A and clinicopathological risk factors or with risk factors alone, 2.15 (1.44–3.21) and 1.61 (1.23–2.10), respectively (Figure 4C).

A forest plot showing the overall multivariate Cox regression analyses of risk for metastasis in cSCC patients with AJCC8 staging system combined with 40-GEP class is presented in Figure 4A.

A forest plot showing multivariate Cox regression analyses of risk for metastasis in cSCC patients with BWH staging system combined with 40-GEP class is presented in Figure 4B.

A forest plot showing the overall multivariate Cox regression analyses of risk for metastasis in cSCC patients with clinicopathologic risk factors combined with 40-GEP class is presented in Figure 4C.

## 4. Discussion

### 4.1. Summary of Evidence

To the best of our knowledge, this is the first meta-analysis to investigate the prognostic value and clinical utility of 40-GEP in the setting of cSCC. The results demonstrate increased accuracy of metastatic risk assessment for high-risk cSCC when using 40-GEP tested both independently and in combination with tumor staging systems as well as with clinicopathologic risk factors.

The current staging systems (AJCC8/BWH) were shown to be insufficient in discerning the patients who are more highly susceptible to the development of metastasis. The inaccurate reporting of different clinicopathological risk factors has led to misclassification and inaccurate staging of high- vs. low-risk tumors among AJCC and BWH classification systems, leading to the under-identification of high-risk tumors associated with high-risk of metastasis and poorer outcomes [8,9]. This was improved with the update of AJCC8 attributed to improved standardization of tumor depth cutoffs and elimination of tumor differentiation [20]. Nonetheless, the PPV of the conventional classification systems remained low and unacceptable, with little aid to the clinician [10,20,21]. PNI is often underreported by less experienced pathologists in smaller medical centers which is a major drawback of the BWH and AJCC8 risk stratification systems. However, PNI was recorded among all patients in all three studies included in the meta-analysis. PNI was associated with increased HR for metastasis of 3.28 vs. 11.61 for PNI and class 2B 40-GEP, respectively, in Ibrahim et al. [12], and 2.63 vs. 9.44 for PNI and class 2B 40-GEP, respectively, in Arron et al. [15]. These results highlight the importance of 40-GEP class 2B over PNI even when PNI is reported by experienced pathologists.

The 40-GEP test was devised as an alternative prognostic tool that quantifies the expression of specific oncogenes that correlated with unfavorable outcomes [12,13,15]. Previous studies have suggested that incorporating GEP into cSCC risk assessment could improve over the current classification systems in discerning the patients who are more highly susceptible to the development of metastasis and would warrant aggressive management with lymph node dissection at the time of tumor excision [12,13,15,16]. The current meta-analysis combined the predictive metastatic risk of 40-GEP with clinicopathologic factors as well as with a tumor staging system (AJCC8/BWH), resulting in improved predictive capacity, particularly in the high-risk 40-GEP class 2B group.

The 40-GEP test results showed statistically significant improvements in several accuracy metrics. The specificity of class 2B from the 40-GEP test was significantly higher (96.52%) than high-stage AJCC8 (84.63%) and BWH (89.97%) classifications. This indicates that a class 2B result has a lower probability of generating a false positive regarding metastasis when compared to AJCC8 and BWH high-stage classifications. Additionally, the sensitivity of a class 2 result (class 2A/2B) from the 40-GEP test (74.45%) was significantly higher than high-stage AJCC8 (37.87%) and BWH (28.33%). Class 2 was considerably more effective in identifying tumors at moderate to high risk of metastasis in comparison to AJCC8 and BWH staging strategies. These results illustrate the potential of molecular tools to improve current risk assessment methods in cSCC.

The current available staging systems yielded a particularly low PPV for the prediction of metastasis: 33% for AJCC8 and 29% for BWH. This is usually explained by the fact that these staging systems do not take into consideration all factors known to be associated with metastasis and the discordance in assessment and reporting among pathologists and surgeons. On the contrary, the 40-GEP is an objective tool with less bias and adds an assessment of known mutations associated with aggressive behavior to the current staging system. Therefore, the PPV of class 2B was significantly higher (55.9%) and surpassed the values associated with high stage according to AJCC8 or BWH staging systems. The combined 40-GEP class 2B with AJCC8 or BWH yielded a more than three-fold improvement in the prediction of metastasis when compared with AJCC8 (HR of 9.98 vs. 2.61) or BWH alone (HR of 8.62 vs. 2.18). Nevertheless, the improvement in prediction was only observed among class 2B group (high risk) and not for class 2A group (intermediate risk). These results suggest that incorporating the 40-GEP high-risk 2B group in the staging of cSCC patients enhances the ability of the current available systems to classify patients into high-risk groups with significantly higher risks of metastasis and worse outcomes.

Other studies have examined the clinical validity of the 40-GEP test and its influence on clinician decision making but were not included in this meta-analysis due to the absence of relevant outcomes. According to Litchman et al., dermatology clinicians were presented with two patient scenarios and instructed to choose management decisions with and without the results of a 40-GEP test and were found to adjust their management plans for high-risk patients (class 2B) by recommending additional adjuvant therapies, nodal imaging, or increasing surveillance and follow-up [22]. The contribution of risk stratification provided by 40-GEP to the consistency of clinician decision-making was also presented in Hooper et al. In the study, clinicians evaluated six real-world cases of cSCC using conventional risk factors and recommended management strategies. They were then asked to re-evaluate the same cases based on the 40-GEP classes and compare their recommendations to the initial assessments. The results showed that the class 2B result had a more significant impact on treatment changes, with an increase in the percentage of clinicians recommending more intense treatments such as sentinel lymph node biopsy (SLNB) or proceeding with adjuvant radiation therapy (ART). A class 2A result had a modest effect on nodal assessment and surveillance imaging. On the other hand, a class 1 result led to de-escalation of treatment intensity. This study provides clear evidence of the clinical utility of the 40-GEP test in guiding clinician decision making for cSCC management [23]. The 40-GEP test has significant potential to assist clinicians to escalate therapy for high-risk patients and de-escalate therapy for the majority of patients with low-risk patients. Farberg et al. have suggested intensifying management in high-risk groups and incorporating 40-GEP within NCCN guideline recommendations [16]. Despite the reported incremental improvements in the prognostic and clinical usefulness of 40-GEP, it has not yet been integrated into the guidelines of the NCCN nor the American Academy of Dermatology [4,24,25].

Several of the genes utilized in the 40-GEP algorithm have been identified in previously reported cSCC cases and were shown to be involved in pathways relevant to cancer [23,26]. However, the role in cancer biology of other genes included in the 40-GEP panel remains unclear and should be examined and evaluated in future preclinical and clinical investigations. Additionally, the literature has proposed several other genes (among them, PLAUR, MMP1, MMP10, MMP13, TIMP4, and VEGFA) as key players in driving metastasis in cSCC, which are not part of the 40-GEP panel [27]. Minaei et al. validated the overexpression of urokinase plasminogen activator receptor (uPAR, encoded by PLAUR) in a larger patient cohort by demonstrating higher uPAR staining intensity in metastatic tumors, supporting its crucial role in promoting metastasis in cSCC and its potential as a therapeutic target. However, further investigations using larger cohort studies and functional studies employing metastasis models of cSCC in vivo are needed to confirm the central role of uPA/R as a biomarker of cSCC metastasis and verify its inclusion in the gene expression profiling panel. Moreover, the identification and incorporation of additional genes associated with poor prognosis to the 40-GEP panel may further improve upon its risk assessment capabilities.

### 4.2. Limitations

There are several limitations of this study that need to be acknowledged. Firstly, the meta-analysis included only three studies, and another two studies that explored GEP integration with staging systems were not suitable due to the absence of relevant data on outcomes. Despite the small size of the meta-analysis, the study design, inclusion criteria, and outcome measurements were uniform, resulting in no interstudy statistical heterogeneity. Another potential limitation is that the samples were archival, and the retrospective studies included may have under-reported high-risk pathological features in the pathological and surgical reports. Lastly, the 40-GEP test was developed to assess the risk of metastasis and not other sequelae, such as recurrence, thus excluding all patients with recurrence from the analysis. Finally, the 40-GEP test was not validated for the risk of local recurrence in any of the studies included.

## 5. Conclusions

The findings of this meta-analysis suggest that the 40-GEP test has substantial prognostic value for high-risk cSCC patients, both independently and when integrated with conventional tumor staging systems and clinicopathological risk factors. However, further large-scale studies that focus on the class 2B group are necessary to better quantify the increase in metastatic risk, guide treatment decisions, modify management strategies, and ultimately, improve patient outcomes and reduce mortality.

## Figures and Tables

**Figure 1 cancers-15-02456-f001:**
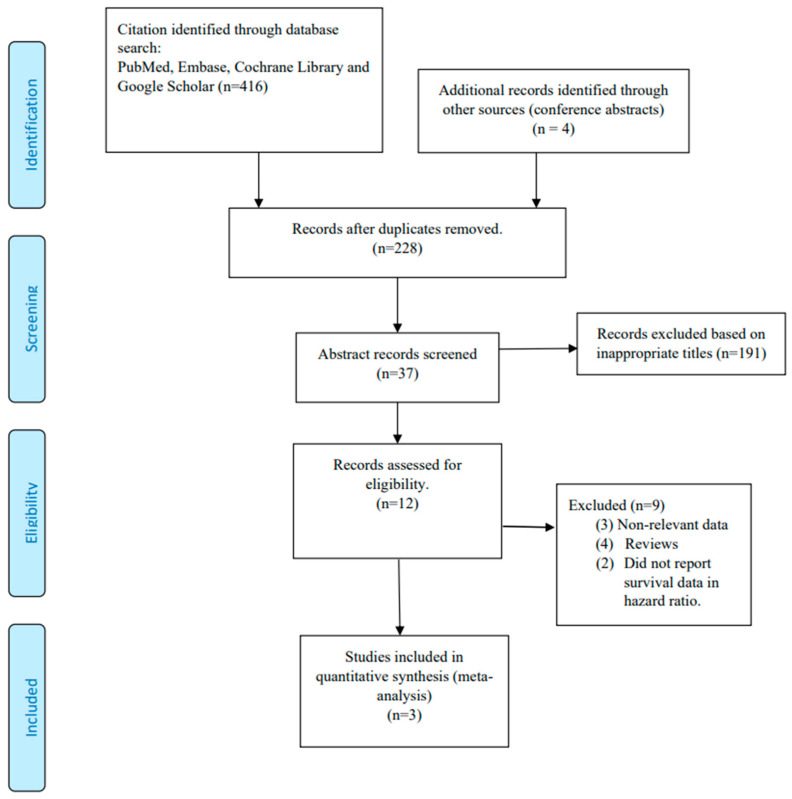
Publication selection and search process.

**Figure 2 cancers-15-02456-f002:**
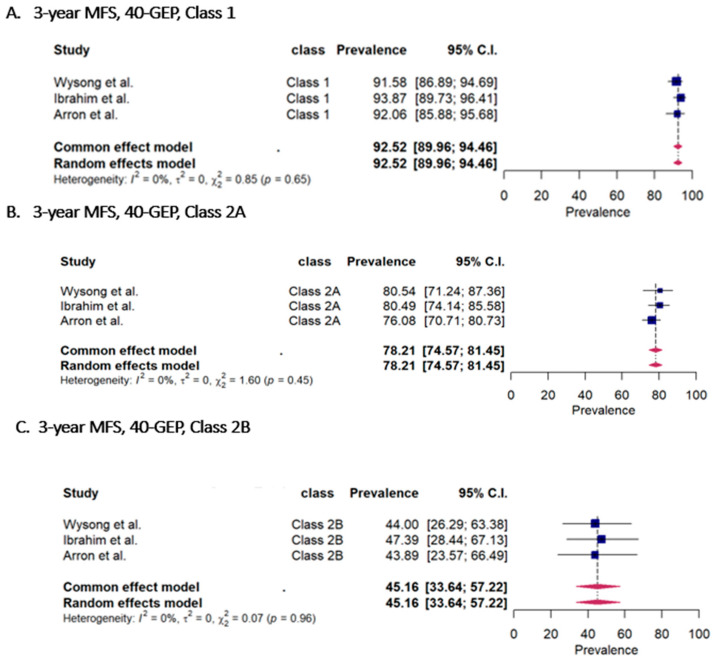
40-GEP as an independent prognostic signature, 3-year metastasis-free survival (MFS) [12,13,15].

**Figure 3 cancers-15-02456-f003:**
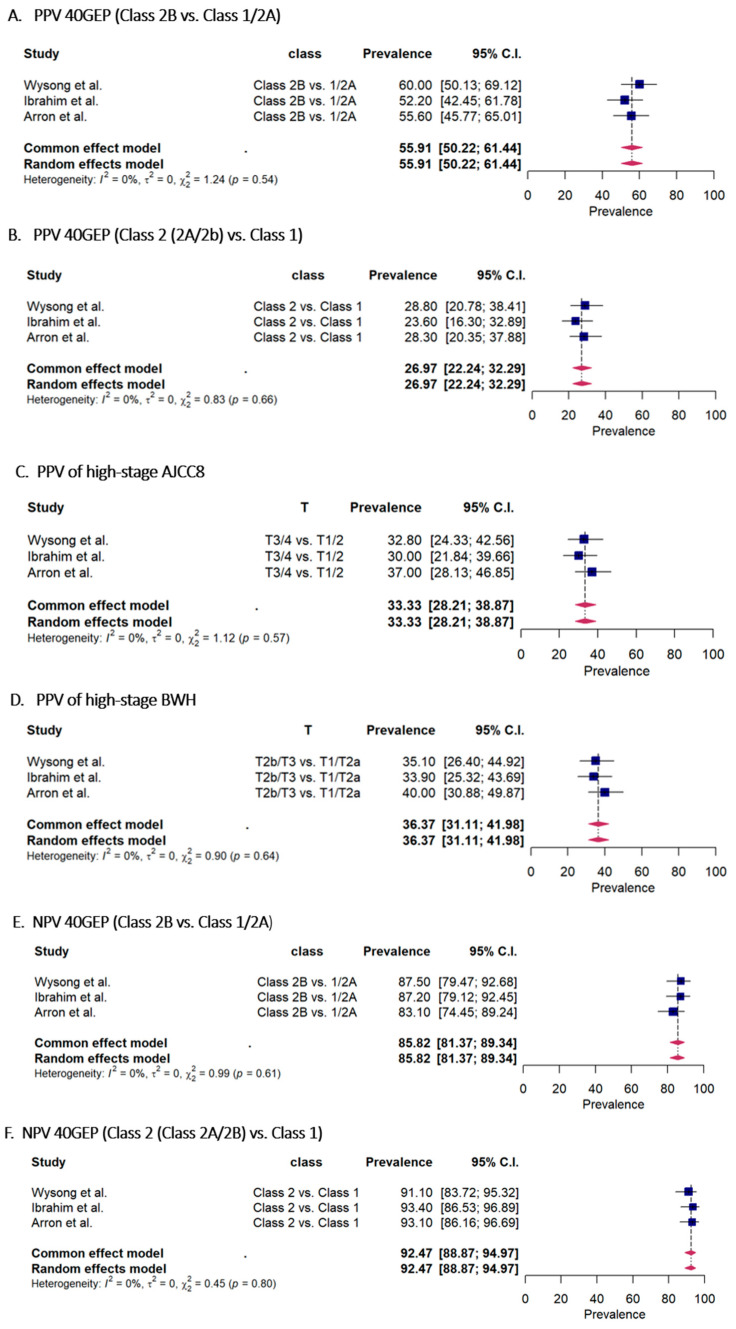
(**A**–**D**) Forest plot of the accuracy metrics; rates of positive predictive value (PPV, **A**–**D**). (**A**) PPV for 40-GEP (class **2B** vs. class 1/**2A**), (**B**) PPV for 40-GEP (class 2 (**2A**/**2B**) vs. class 1), (**C**) PPV for AJCC8 (T3/T4 vs. T1/T2), (**D**) PPV for BWH (T2b/T3 vs. T1/T2a). AJCC8, American Joint Committee on Cancer Cancer Staging Manual, eighth edition; BWH, Brigham and Women’s Hospital; GEP, gene expression profile [12,13,15]. (**E**–**H**) Forest plot of the accuracy metrics; rates of negative predictive value (NPV, **E**–**H**). (**E**) NPV for 40-GEP (class **2B** vs. class 1/**2A**), (**F**) NPV for 40-GEP (class 2 (**2A**/**2B**) vs. class 1), (**G**) NPV for AJCC8 (T3/T4 vs. T1/T2) and (**H**) NPV for BWH (T2b/T3 vs. T1/T2a). AJCC8, American Joint Committee on Cancer Cancer Staging Manual, eighth edition; BWH, Brigham and Women’s Hospital; GEP, gene expression profile [12,13,15].

**Figure 4 cancers-15-02456-f004:**
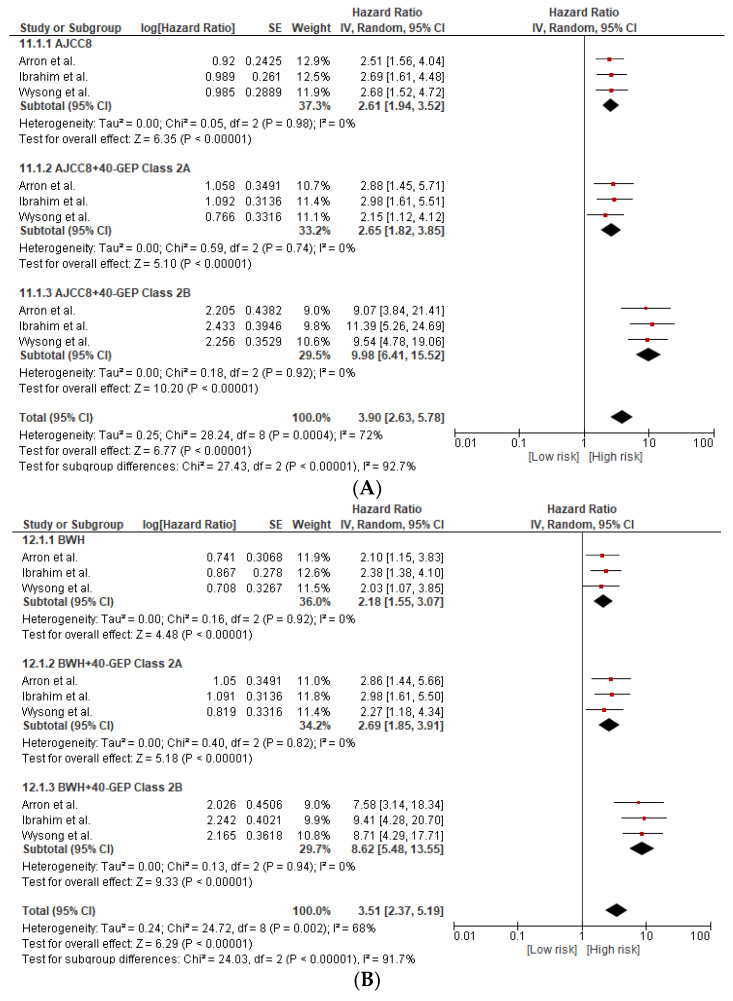
(**A**). Subgroup analysis based on randomization for AJCC8 staging and AJCC8 combined with either 40-GEP class 2A or class 2B. The pooled HRs were 2.61 (95%CI 1.94–3.52; *p* < 0.00001), 2.65 (95%CI 1.82–3.85; *p* < 0.00001), and 9.98 (95%CI 6.41–15.52; *p* < 0.00001). By definition, HR > 1 implied a high risk of disease metastasis. AJCC8, American Joint Committee on Cancer Cancer Staging Manual, eighth edition; GEP, gene expression profile; IV, inverse variance; CI, confidence interval; HR, hazard ratio; p, *p*-value [12,13,15]. (**B**) Subgroup analysis based on randomization for BWH staging and BWH combined with either 40-GEP class 2A or class 2B. The pooled HRs were 2.18 (95%CI 1.55–3.07; *p* < 0.00001), 2.69 (95%CI 1.85–3.91; *p* < 0.00001), and 8.62 (95%CI 5.48–13.55; *p* < 0.00001). By definition, HR > 1 implied a high risk of disease metastasis. BWH, Brigham and Women’s Hospital; GEP, gene expression profile; IV, inverse variance; CI, confidence interval; HR, hazard ratio [12,13,15]. (**C**). Subgroup analysis based on randomization for clinicopathological risk factors combined with either 40-GEP class 2A or class 2B. The pooled HRs were 1.61 (95%CI 1.23–2.10; *p* < 0.0005), 2.15 (95%CI 1.44–3.21; *p* < 0.0002), and 6.66 (95%CI 4.04–10.96; *p* < 0.00001). By definition, HR > 1 implied a high risk of disease metastasis. GEP, gene expression profile; IV, inverse variance; CI, confidence interval; HR, hazard ratio [12,13,15].

**Table 1 cancers-15-02456-t001:** Characteristics of the included studies.

Study	Design, Population	Age, y, Median	Male (%)	Immunosuppression (%)	Location in Head and Neck (%)	PNI (%) Present (≥0.1 mm)	Invasion Beyond Subcutaneous Fat (%)	AJCC8 (%)	BWH (%)
Wysong et al. (2020) [13]	Patients with ≥1 high-risk features as defined by NCCN guidelines or by AJCC or BWH staging as being greater than T1. (*n* = 321)	70 (34–95)	73.2%	23.7%	66.7%	2.2%	13.4%	T1 (62.6%) T2 (18.4%) T3 (16.8%) T4 (2.2%)	T1 (57.9%) T2a (30.5%) T2b (9.4%) T3 (2.2%)
Ibrahim et al. (2021) [12]	Patients with ≥1 high-risk features as defined by NCCN guidelines or by AJCC or BWH staging as being greater than T1. (*n* = 420)	71 (34–95)	73.3%	24.5%	66.2%	1.7%	12.1%	NR	NR
Arron et al. (2022) [15]	Patients with ≥1 high-risk features as defined by NCCN guidelines or by AJCC or BWH staging as being greater than T1. (*n* = 278)	71 (34–95)	82.4%	22.7%	100%	2.5%	13.7%	T1 (57.6%) T2 (23.0%) T3 (15.8%) T4 (3.6%)	T1 (50.7%) T2a (34.9%) T2b (10.8%) T3 (3.6%)

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
