# Peer review of "The Prognostic Value and Clinical Utility of the 40-Gene Expression Profile (40-GEP) Test in Cutaneous Squamous Cell Carcinoma: Systematic Review and Meta-Analysis"

_cancers, 2023, doi:10.3390/cancers15092456_

Round 1

Reviewer 1 Report

Authors review and compare the 40 gene expression test, the ajcc8 and bwh classification for prognosis prediction in cSCC. The manuscript is well written and structured.

In the introduction it is stated that nrve invasion is often not documented hampering classification. This is one of the reasons the 40 GEP test is important for predicting prognosis. Yet, in the comaparisons and combinations, authors include the same classification system. I am a bit confused, is the 40 gene GEP compared to fully completed descriptions (with nerve inervation described) or not? If not, would the advice be that we should do better and describe nerve inervation or should be always include the 40 GEP test? Please elaborate.

Is metastasis free survival in the same way across studies?

The analysis is focussed on averages/groups. Does the 40 gene GEP test have outlyers? or subgroups? Why not show a Kaplan-Meider plot?

Do the results of the meta-analysis differ from those of he incorporated studies?

Author Response

We thank you and all reviewers once again for considering our manuscript and for the comprehensive evaluation of the manuscript.

Herein are our responses to the reviewers’ comments:

Reviewer 1:Comments and Suggestions for Authors

Authors review and compare the 40 gene expression test, the AJCC-8th, and BWH classification for prognosis prediction in cSCC. The manuscript is well-written and structured.

Point 1: In the introduction, it is stated that nerve invasion is often not documented hampering classification. This is one of the reasons the 40 GEP test is important for predicting prognosis. Yet, in the comparisons and combinations, authors include the same classification system. I am a bit confused, are the 40 genes GEP compared to fully completed descriptions (with nerve innervation described) or not? If not, would the advice be that we should do better and describe nerve innervation or should we always include the 40 GEP test? Please elaborate.

Response 1: Thank you for this comment. We acknowledged that inconsistent reporting of perineural invasion (PNI) by pathologists, often in smaller center and non-cancer centers, is a major drawback of the BWH and AJCC-8th risk stratification systems. However, PNI was recorded among all patients in all three studies presented in our meta-analysis. PNI was associated with increased multivariate HR for metastasis of 3.28 vs 11.61 for PNI and class 2B 40-GEP, respectively, in Ibrahim et al, and 2.63 vs 9.44 for PNI and class 2B 40-GEP, respectively, in Arron et al. These results only highlight the importance of 40-GEP class 2B over PNI even when PNI was fully evaluated and reported by experienced pathologists.

Point 2: Is metastasis-free survival in the same way across studies?

Response 2: All the included studies have reported a 3-year metastasis-free survival (MFS) in the same manner. Kaplan–Meier and log-rank tests were similarly performed and reported for MFS analyses in each of the studies.

Point 3: The analysis is focused on averages/groups. Does the 40-gene GEP test have outliers? or subgroups? Why not show a Kaplan-Meier plot?

Response 3:

  • The Kaplan-Meier results of each study were incorporated and pooled into forest plots to calculate MFS percentages.
  • 40-gene GEP has three subgroups: group 1, group 2A, and group 2B. Only group 2B according to the individual studies and according to the meta-analysis shows a stronger association with metastasis when compared to the current staging systems as presented in the meta-analysis. We meta-analyzed individual hazard ratios (HRs) with corresponding 95% CIs from the most adjusted multivariable model, using the generic inverse variance approach with random-effects modeling. Pooled summary estimates were reported as HR
  • As the sensitivity and specificity of the 40-gene GEP are not 100%, outliers will be found with false positive and false negative results. The specificity of a Class 2B the 40-GEP test was 96.52% which is significantly higher compared to high-stage AJCC8 (84.63%) and BWH (89.97%) classifications. Class 2B result has a lower probability of generating a false positive regarding metastasis when compared to AJCC8 and BWH high-stage classifications. Similarly, the sensitivity of a Class 2 result (Class 2A/2B) from the 40-GEP test (74.45%) was significantly higher than high-stage AJCC8 (37.87%) and BWH (28.33%).

Point 4: Do the results of the meta-analysis differ from those of the incorporated studies?

Response 4: The was no heterogenicity between the three studies with similar results among all studies supporting the superiority of 40-GEP class 2 and class-2B in particular which empowers the results of the meta-analysis. When pooling the results of the studies, our results support the results of the studies.

Reviewer 2 Report

Razan Masarwy and co-authors present a review focused on the utility of the 40-Gene Expression Profile for the prognostic of cutaneous squamous cell carcinoma. 

The protocol used for the meta-analysis is adequate and the information analyzed is clearly presented. However, as the authors stated in the conclusions, 3 articles are not enough to confirm the utility of the 40-gene expression profile. In this sense, the number of patients in the 3 papers is more than 1000 which increases the robustness of the study. 

1. References need to be before the punctuation sign. 

2. It is appropriate to register the meta-analysis protocol in PROSPERO. This could be very useful for further studies. 

Author Response

We thank you and all reviewers once again for considering our manuscript and for the comprehensive evaluation of the manuscript.

Herein are our responses to the reviewers’ comments:

Reviewer 2:

Razan Masarwy and co-authors present a review focused on the utility of the 40-Gene Expression Profile for the prognostic of cutaneous squamous cell carcinoma. 

Point 1: The protocol used for the meta-analysis is adequate and the information analyzed is clearly presented. However, as the authors stated in the conclusions, 3 articles are not enough to confirm the utility of the 40-gene expression profile. In this sense, the number of patients in the 3 papers is more than 1000 which increases the robustness of the study. 

Response 1: We appreciate your thorough comment. We agree with the reviewer that 3 studies are on the lower side when conducting a meta-analysis. However, as reviewer 2 mentioned, more than 1000 patients were included, and all three studies were very similarly conducted and were very homogeneous which empowers and increased the robustness of the meta-analysis.

Point 2: References need to be before the punctuation sign. 

Response 2: Thank you for this remark. We changed the reference numbers to before the punctuation sign.

Point 3: It is appropriate to register the meta-analysis protocol in PROSPERO. This could be very useful for further studies. 

Response 3: Thank you for your comment, the meta-analysis was registered in the International Prospective Register of Systematic Reviews registry (PROSPERO) and we attached the registration number in the methods section – first paragraph. [CRD42023400695].
